# What factors impact student performance in introductory physics?

**Eric Burkholder**[1]*, **Lena Blackmon**[2], **Carl Wieman**[1,3]

**1** Department of Physics, Stanford University, Stanford, California, United States of America, **2** Department of Applied Physics, Stanford University, Stanford, California, United States of America, **3** Graduate School of Education, Stanford University, Stanford, California, United States of America

* eburkhol@stanford.edu

**Data Availability Statement:** Data cannot be shared publicly as a condition of IRB approval. Data may be available from the Stanford University Student Data Oversight Committee (contact cjpotter@stanford.edu).

## Abstract

In a previous study, we found that students' incoming preparation in physics—crudely measured by concept inventory prescores and math SAT or ACT scores—explains 34% of the variation in Physics 1 final exam scores at Stanford University. In this study, we sought to understand the large variation in exam scores not explained by these measures of incoming preparation. Why are some students' successful in physics 1 independent of their preparation? To answer this question, we interviewed 34 students with particularly low concept inventory prescores and math SAT/ACT scores about their experiences in the course. We unexpectedly found a set of common practices and attitudes. We found that students' use of instructional resources had relatively little impact on course performance, while student characteristics, student attitudes, and students' interactions outside the classroom all had a more substantial impact on course performance. These results offer some guidance as to how instructors might help all students succeed in introductory physics courses.

## Introduction

Instructors of introductory physics courses frequently observe large variations in student performance in their courses, despite instructors providing the same instruction to all students. Much work in Physics Education Research has been devoted to improving average student performance in introductory courses, but only recently have researchers and instructors begun to ask, what is the cause of these variations in student outcomes and how can they be addressed to help all students be successful? A primary concern in this regard has been differences in course performance between male and female students, URM and majority students, and first-generation and continuing generation students [1–6].

In a previous paper, we analyzed data from the introductory Mechanics course, Physics 1, at three large research universities and found that differences in average student performance (as measured by final exam scores) between subpopulations of students could all be explained by average differences in students' math SAT (or ACT) score and physics concept inventory prescore [6]. Moreover, we found that the *only* statistically significant predictors of course performance, of the data available to us, were these two measures of students' incoming preparation, *not* demographic characteristics. These measures explained 20–34% of the variation in final exam scores [6].

**Funding:** The authors received do specific funding for this work.

**Competing interests:** The authors have declared that no competing interests exist.

In that paper, we used multiple linear regression (MLR) to quantify the relationship between these measures of student preparation and course performance. Specifically, we tested the following relationship between final exam scores and our two admittedly crude measures of incoming preparation:

$$z_{Final} = b_0 + b_{SAT}z_{SAT} + b_{CI}z_{CI} + b_{Int}z_{SAT} \times z_{CI} + \epsilon \qquad (1)$$

where $z_{Final}$ were students' final exam scores, $z_{SAT}$ were students' math SAT or ACT percentile scores, and $z_{CI}$ were their concept inventory prescores. All continuous variables were converted to z-scores and the regression coefficients $b_i$ were estimated by minimizing the regression error $\epsilon^2$, as is typical in MLR. For students at Stanford University ("HSWC" in Ref. [6]), we found $b_{SAT} = 0.35\pm0.05$ and $b_{CI} = 0.34\pm0.05$. This means that, for every standard deviation increase in SAT scores there was a 0.35 standard deviation increase in final exam scores, and for every standard deviation increase in concept inventory pre-scores there was a 0.34 standard deviation increase in final exam scores. We found that interaction terms were not statistically significant and neither were a number of other variables we tested including demographic status and some social psychological factors, and so they were not included in the final model in Table IV of that paper. The linear combination of concept inventory prescores and math SAT or ACT scores explained 34% of the variance in students' final exam scores. While 34% of total variance explained is quite large for an education study, it is far less than 100%. An imperative research question is: what explains the remaining variation—what are the factors contained in the regression error $\epsilon$ in Eq 1?—and are these factors that can be impacted by improved instruction?

Some previous studies have attempted to determine the effects of other variables on course performance. For example, Ballen et al. [7] showed that, for female students in introductory biology, test anxiety had a direct negative impact on course performance while interest in the material had a positive impact on course performance. Salehi et al. showed that this negative effect of test anxiety persists across scientific disciplines, and throughout students' undergraduate careers [8]. Others have looked at the impact of social psychological factors such as self-efficacy [9], but we found no effect of these variables in Ref. [6]. In a more general context, Plant et al. found that students' study habits were predictive of cumulative GPA, but that the amount of time spent studying did not correlate with GPA [10].

As no other variables readily available to us were statistically significant predictors of course performance, we decided to interview students to identify other "hidden" variables that may explain this variation in performance. Specifically, we interviewed 34 students from Stanford University with relatively low levels of incoming preparation (as defined by our two variables) but with a variety of course outcomes. We asked them a series of questions about their background, study habits for the class, and experience in the course. We focused on students with low levels of preparation because we were particularly interested in discovering what factors allow some of these students to succeed despite their apparent low levels of preparation. We focused on students' backgrounds and experiences during the course, because these seemed the most likely to be related to their course performance, and because we had found that social psychological factors were not predictive of course performance in this population.

We did not form initial hypotheses for what we might learn. First, we, as researchers, have been successful in physics, and so we thought it unreliable to judge the experiences of these students based on our own experiences in introductory physics. Second, in our past efforts, as well as efforts of others we know about, getting detailed information from poorly performing students has been notoriously difficult, and so we had very little prior information to guide hypotheses. Not surprisingly, such students are not eager to talk about their experiences in a

course they found difficult and in which they performed poorly [11, 12]. We were successful in recruiting this relatively large sample of such students for this study only by using very persistent recruiting and inducement efforts.

In the absence of prior data, it seemed possible that the "hidden variables" were a large set of idiosyncratic variables with little in common from one student to the next, and largely outside the context of the course and beyond the control of an individual instructor. Alternatively, it seemed possible that the missing variables were issues common to many students and related to the course elements, students' preparation, or social-educational aspects (such as peer support networks) and hence might be addressed through instruction. As discussed in detail below, we found the latter was largely true, with students mentioning a total of only 7 different attitudes and 15 different practices, 10 of which stood out as being mentioned by many students.

This small number of practices identified may be somewhat unique to Stanford students. They are relatively free from some factors that make school difficult for many college students, such as immediate economic, living, and transportation problems, and the burden of family responsibilities such as caring for children or relatives. Even so, the practices we identified in this population are likely to still be relevant to many students, though they may not be the whole story.

## Methods

Our sample population was 394 Physics 1 students for whom we had Force and Motion Conceptual Evaluation (FMCE) prescores and postscores [13], math SAT or ACT scores, final exam grades, and demographic variables at Stanford University. These quantitative data were collected during the 2017–2018 academic year; all students were enrolled in one section taught by a single lecturer. This is a single year subset of the sample population used in Ref. [6]. From these 394 students, we chose 50 students whose FMCE prescores were below the 30th percentile of the class distribution; 25 of these students had taken an additional companion course designed for students without previous experience in physics. These 50 students were selected to represent FMCE postscores that covered a wide range of the score distribution, but they were selected without knowledge of their math SAT or ACT scores, final exam scores, or demographic variables. We selected these students so that we might find out what factors help some students succeed in physics despite low levels of incoming preparation. We used final exam scores as a measure of course performance because it is a measure which is more consistent across student populations. Course grades are comprised of many components, such as participation, and the breakdown of course grades varies widely by instructor and course. Final exams, on the other hand, are consistently high-stakes summative assessments that measure similar content across courses and institutions, and for this particular course are a large factor in determining the course grade.

The course we studied was a typical calculus-based introductory physics course for engineering and science majors. It covered kinematics, conservation of energy, conservation of momentum, conservation of angular momentum, force, torque, and uniform circular motion. The course consisted of traditional one-hour lectures three times per week with limited use of Clicker questions. There was also a recitation section that met once per week in which students usually worked on problems from Tutorials in Introductory Physics.Some students also elected to take a 1-credit companion course which met for two hours once per week. The companion course was focused on teaching problem-solving and exam skills. Students would work in small groups to complete problems while instructors circled the room to provide feedback, practice taking timed exams, or students would listen to instructors explain particularly difficult concepts.

These 50 students received targeted emails asking them to participate in a study aimed at improving the undergraduate physics experience for students at Stanford and offering them $40 for an hour of their time. Nineteen students responded to the initial email (11 of whom were in the companion course), and three students responded to a reminder email (one of whom was in the companion course). Another ten students responded after receiving additional individual emails, five of whom were in the companion course. Other attempts were made to contact the remaining 18 students, including mailed solicitation letters and contacts through third parties such as diversity offices or academic administrators in the students' home departments. This resulted in two more students (neither of whom were in the companion course) responding. We were unusually persistent in recruiting students because we wanted to make sure that our respondents were representative of the whole sample we had chosen in terms of prior preparation, course performance, and demographics (see below). We note that no student was contacted more than four times over a period of 5 months to avoid harassing the students.

In all, 34 of the 50 students were interviewed. Between the students who responded and those who did not, there were no statistically significant differences in demographics, FMCE postscores, exam scores, or math SAT or ACT scores (see Table 1). Responders had higher FMCE prescores, but this is not statistically significant if p-values are corrected for multiple comparisons using the Bonferroni procedure [14]. We note that the sample of students we interviewed is unusually diverse, but this is reflective of the population of students with this range of FMCE prescores.

The interviews were conducted between 6 months and one year after students had completed Physics 1; they ranged from 20–60 minutes in length and were transcribed by the first author. The first author conducted the first 19 interviews individually, but the interviews of students who did not respond to intial email requests and follow-ups were interviewed by the first and second authors jointly. The interviews were semi-structured and based on the 12 guiding questions given in Table 2. The questions were not always asked in this order or with the precise wording given, but all topics were covered in each interview. Further probing questions were asked to clarify earlier responses and/or further explore issues that had been raised. The interviews were presented to the students as a way to improve instruction in the introductory physics course.

We cannot rule out the possibility that some important aspects of students' experiences were omitted because we did not ask directly about them or because students did not remember or were uncomfortable discussing them. However, all students provided detailed accounts of their physics 1 experiences, and it seems plausible that most aspects that students did not mention were less important. We also cannot rule out the possibility that students' responses

**Table 1. Descriptive statistics for students who did and did not respond to requests for interviews.** Differences were tested by the Mann-Whitney U-test, and p-values have not been corrected for multiple comparisons.

| Variable | Responders | Non-responders | Difference |
|---|---|---|---|
| Gender (F = 1) | 65% (s.d. = 0.49) | 56% (s.d. = 0.51) | U = 249, p = 0.58 |
| URM | 59% (s.d. = 0.50) | 44% (s.d. = 0.51) | U = 313, p = 0.33 |
| FG | 35% (s.d. = 0.49) | 25% (s.d. = 0.45) | U = 300, p = 0.48 |
| Companion Course | 50% (s.d. = 0.51) | 50% (s.d. = 0.52) | U = 272, p = 1 |
| FMCE prescore | 12 (s.d. = 4.6) | 18 (s.d. = 8.9) | U = 146.5, p = 0.0091 |
| FMCE postscore | 28 (s.d. = 12) | 34 (s.d. = 10) | U = 194.5, p = 0.11 |
| SAT or ACT math | 690 (s.d. = 50) | 720 (s.d. = 60) | U = 195, p = 0.11 |
| Final Exam | 65 (s.d. = 19) | 73 (s.d. = 15) | U = 215, p = 0.24 |

**Table 2. Guiding and follow-up questions for student interviews.** Not all questions were asked exactly as written.

| Question | Typical Follow-ups |
|---|---|
| 1. To start off, can you tell me a little bit about yourself, like where you're from and where you went to high school? | a. Did you take physics or advanced math?<br>b. What was that like? |
| 2. How long have you been at Stanford? | |
| 3. What's your major (or intended major)? | a. Why did you choose that?<br>b. Do you know what you want to do after Stanford?<br>c. Do you do any extracurriculars here? |
| 4. Can you tell me about your experience in physics 1? | a. What did you think of the sections?<br>b. What did you think of the lectures?<br>c. What were your interactions like with other students? |
| 5. Did you take the companion course? | a. What did you think of that?<br>b. What did you think about the group-work aspect of the companion course? |
| 6. Can you tell me about what you would do to study for the class? | a. How much time would you spend on the class in a given week?<br>b. Did you use any external resources?<br>c. Did you ever read the book before class?<br>d. Did you study (or do problem sets) with others?<br>e. Would you review solutions or re-work homework problems when studying (if mentioned)?<br>f. Did you do any extra practice problems? |
| 7. Did you take physics 2 or [engineering statics]? Do you feel like physics 1 prepared you for that? | |
| 8. What were the best aspects of the course for you? | |
| 9. On the other side of that, what were the worst? | |
| 10. How did physics 1 compare to other STEM courses you've taken at Stanford? | a. Did you take chemistry 1? If so, how did it compare? |
| 11. If you had to go back and give yourself some advice before taking this class, what do you think you would say? | |
| 12. What did you think about physics before taking physics 1 versus coming out of the class? | a. Did it influence your career/educational plans? |

were skewed by other courses they were currently taking (e.g. contrasting their current and prior experiences). We expect the students may have been comfortable with the interviewers because one of them was a postdoc new to Stanford, and often framed his line of questioning as a way of finding out what the student experience at Stanford was like, and the second interviewer was a recent undergraduate student and woman of color, who could empathize with students' experiences. This study was approved by the Stanford University Institutional Review Board, protocol number IRB-48006. All participants gave written consent for anonymous quotes and aggregated data to be used.

We first used qualitative methods (thematic analysis of the extended interviews) to determine what issues were important. This type of exploratory analysis is best suited to this research, particularly since we started with no hypotheses about what issues were important and how diverse the set of issues might be. We then took a semi-quantitative approach to identify the prevalence of these identified issues in a large number of interviews.

We used thematic analysis to analyze the qualitative data in terms of issues students raised as relevant to their experience and performance in the course. The first and second authors coded each interview independently in batches of 5–7 interviews (batches were ordered chronologically by interview date). After each batch had been coded, the two coders met to discuss what issues they had noticed and what they thought were the most important features of each

issue. The definitions of each of the identified issues were refined after coding of further interviews, and some additional issues were identified. After all interviews had been independently coded, a final coding scheme was established with 22 defined issues—seven of which were student attitudes, and 15 of which were student practices. Both raters then recoded all interviews according to this final coding scheme. This provided good interrater reliability, Cohen's $\kappa$ = 0.812) [15]. The quantitative analysis below is carried out using the coding of all interviews by the second author.

Examples of codes are:

- Worked collaboratively with other students to do homework

- Read text before coming to class

- Did extra (beyond those assigned or suggested in class) practice problems in preparation for exam

- Lacking confidence in physics

For "semi-quantitative" analysis, we calculated students' predicted final exam scores based on their prior preparation (see Eq 1). We then identified whether a student performed better than predicted (N = 15, where N is the number of students), or worse than predicted (N = 19). We then computed the fraction of students in each category who mentioned a particular code to find which particular codes corresponded to higher performance. As the number of participants in each category is small and this study is primarily qualitative, we do not conduct any inferential statistical tests. We draw conclusions based upon the fraction of students in each category who mentioned a particular code.

## Results

We coded 22 issues in our thematic analysis, 7 of which were items that reflected students' attitudes toward the course. The attitudes with brief descriptions are listed in Table 3 along with the number of students who mentioned them. We recorded the overall change in students' attitude towards physics during the course (line 1 in Table 3). A small number of students exhibited explicit growth (2) or fixed mindsets (3) toward intelligence in their interviews, and a similarly small number exhibited grit (4). More students exhibited a lack of confidence in their ability to do physics (5) and a large number felt that the course moved too fast (6) or that there was a disconnect between the types of examples they saw in class versus on the homework (7). We did not conduct a detailed quantitative analysis of any of these attitudes because they were either mentioned by a small number of students (less than 10), or because it was difficulty to determine wether the effects were causes or effects of student performance in the course.

From our thematic analysis, we coded 15 different practices (Table 3). Five of the 15 practices were mentioned by four or fewer students: meeting one-on-one with the course staff (#17 in Table 3), (#21) participating in NCAA athletics, (#12) leading group discussions in the companion course, (#14) struggling to keep up in the companion course working groups, and (#13) feeling like their entire companion course working group was not keeping pace with the rest of the class. Because these were not widespread amongst the interviewees, we did not include them in the quantitative analyses that follow. The remaining 10 practices were each mentioned by 10 or more respondents, thus indicating they are seen as important by a large fraction of students. As we are comparing the populations of students who scored higher than predicted by our incoming preparation model (Eq 1) and lower than predicted by incoming preparation model, respectively, on the final exam, it is worthwhile to ask what additional

**Table 3. List of issues that arose from qualitative analysis of student interviews.** Percentages shown are the fraction of the students in the respective course performance category that listed that attitude or practice. Note that the in-class interactions 12–15 are based only on the students who were in the companion course (N = 17); the remaining issues are based on the entire sample. The * indicates a practice mentioned by a very small number of students and we caution the reader that inference is quite limited.

| Category | No. | Code | Better than Predicted (N = 15) | Worse than Predicted (N = 19) | Description |
|---|---|---|---|---|---|
| Attitudes | 1. | Attitude Change | 13% Neg. 27% Neu. 60% Pos. | 11% Neg. 42% Neu. 47% Pos. | Whether a student's attitude toward physics changed as a result of the course |
| | 2. | Growth Mindset | 20% | 11% | Evidence of a growth mindset toward intelligence |
| | 3. | Fixed Mindset | 6.7% | 5.3% | Evidence of a fixed mindset toward intelligence |
| | 4. | Grit | 20% | 11% | Exhibiting grit–perseverance in the face of adversity |
| | 5. | Lack of Confidence | 20% | 32% | Expressing the feeling that other students in the class are ahead of them |
| | 6. | Pacing | 53% | 58% | Said the course moved too fast |
| | 7. | Disconnect | 40% | 26% | Said there was a disconnect between in-class examples and homework/exam problems |
| Practices | 8. | Intentional | 67% | 79% | Intentionally interacted with other students outside of class |
| | 9. | Spurious | 27% | 42% | Interacted with other students outside of class by chance |
| | 10. | Productive | 60% | 63% | Had out-of-class interactions that were generally productive for learning |
| | 11. | Unproductive | 27% | 47% | Had out-of-class interactions that were generally unproductive for learning |
| | 12.* | Leader | 25% | 0% | Lead the discussion in in-class working groups |
| | 13.* | No Directionality | 0% | 10% | Was in a group where no student knew what to do |
| | 14.* | Left Behind | 38% | 30% | Struggled to keep up with in-class groups |
| | 15. | Peer Instruction | 50% | 60% | Worked together productively with in-class groups |
| | 16. | External Resources | 53% | 58% | Used external/online resources to study |
| | 17.* | One on One | 0% | 10% | Met one-on-one with course staff |
| | 18. | Extra Practice | 47% | 58% | Did extra practice textbook problems |
| | 19. | Read Ahead | 33% | 26% | Read the textbook before class |
| | 20. | Gap Year | 33% | 53% | Waited until after freshman year to take Physics 1 |
| | 21.* | Athlete | 0% | 21% | Participated in NCAA athletics |
| | 22. | Extracurriculars | 67% | 89% | Participated in extensive extracurricular activities |

factors might impact exam performance that we are missing. Although several have previously been identified such as stereotype threat and others that cause high levels of exam-related stress [8], while issues about the exams were noted during these interviews, exam-related stress was not a mentioned as a significant issue.

There were four important practices related to out-of-class social interactions: unproductive interactions (#11 in Table 3), productive interactions (#10), intentional interactions (#8), and spurious interactions (#9). One practice related to in-class social interactions was raised: students working together cooperatively in in-class groups (#15). We noted three practices related to instructional resources: students' use of external resources to aid their study (#16), reading the textbook before class (#19), and doing extra practice problems from the textbook (#18). There were two practices related to student characteristics & extracurriculars: whether a student chose to wait until after their freshman year to take Physics 1 (#20) and whether the student was significantly involved in extracurricular activities (#22). We provide detailed definitions and supporting quotes to define each of these practices below.

### Out-of-class social interactions

Reports of out-of-class student-student interactions were nearly universal (*N = 32*). These interactions appear to be an integral part of the student experience in Physics 1. Here we characterized such interactions along two dimensions according to how they were described in interviews: spurious vs. intentional, and productive vs. unproductive.

### Intentional vs. spurious

Intentional interactions (#8 in Table 3) reflected a conscious effort to do homework or study with one or more students regularly:

> *"Yeah, so I had. . .well there was a big study group in my dorm that would get together and try to do the problem sets together. Um. . .there was. . .also I had friends that I made through the class or through other activities that I would also consult for help. I would work on different groups with different people. . .I had several different study groups I would say."*

Spurious interactions (# 9 in Table 3) were happenstance, such as discussing problems with other students who were at office hours at the same time, or encountering a classmate in a dorm or other public area while studying, e.g.

> *"I had many office-hours friends. But, no, I would sometimes, in the dorm, there was someone, but it was harder to do because a lot of my friends had taken it the freshman sequence. So, really, outside of office hours, there wasn't a whole lot of collaboration because it was more difficult."*

Both spurious and intentional interactions were more common among students who performed worse than predicted by our regression model (42% and 79%, respectively) than among students who performed better than predicted (27% and 67%, respectively). Intentional interactions were more prevalent overall.

### Productive vs. unproductive

The effect of the productivity of out-of-class interactions is larger. Productive interactions (# 10 in Table 3) were ones that students characterized as productively struggling with the material by working with peers whose knowledge of physics was similar to their own, or where students had the opportunity to teach the material to others. Unproductive interactions (#11 in Table 3) were those where students were primarily studying with one other person who was far more prepared than they were (the complement to the students who had opportunities to teach others) or where students indicated that they relied too heavily on peers and did not indicate that they ever spent individual time learning the material:

> *"So I feel like that's probably the main thing I would have told myself. And, to just go to office hours even when, like. . .sometimes I would want to go with other people, and at the start it was like, "They don't want to go. Fine, I guess I won't go." So I would tell myself, like, "You know what? Just go, it doesn't matter other people don't want to." Just go on your own because that was also a main part of helping me."*

Productive Interactions were more common than unproductive interactions, with 60% of students who performed better than predicted and 63% of students who performed worse than predicted reporting such interactions. Though unproductive interactions were less common

overall, they were more prevalent among students who performed worse than predicted (47% vs. 27% of students who perform better than predicted).

## In-class social interactions

An integral part of the recitation sections and companion course sessions involved students working in groups of 3–4 on a series of physics problems related to the material covered in the lecture that week. In the recitation section, groups were ad-hoc, and members varied from week-to-week (mostly due to inconsistent attendance, according to the students interviewed) and the collaborations were unstructured and often minimal. In the companion course, group work was a major part of the class and well-structured. Groups were ad-hoc the first day of class but were then reformed based on results from the FMCE pretest to be made up of students with similar scores, and the composition was unchanged for the remainder of the term. The group composition also ensured that there were no groups with only one student from a demographic group underrepresented in physics—relatively easy and inconspicuous to do in this population, as this represented the majority of the students. We analyze the in-class interactions results from students in the companion course only, as group membership and type of interactions in the regular recitation sections was too dynamic to yield a useful signal.

In the interviews, students often expressed whether they perceived their groups were functioning or not. Well functioning groups (#15 in Table 3) were denoted by students feeling like they learned something from working with their peers:

> "Um, and, oftentimes those people could be like my teachers, or they could help me, like, further understand the problem better, or they'd have like a unique approach, um, and maybe, like. . .oftentimes going back and forth we were able to solve problems through. So I really enjoyed that part of the, uh. . .of that, like, component of the class."

or when students described behaviors that are known to be beneficial for learning, such as explaining concepts to one another:

> "but sometimes when you get an explanation from another student it'll make a lot more sense. And that's a lot of what happened in class—where we explained it to each other. So, like, if. . .I feel like it depended on the group that you were in at least."

We call this effect "Peer Instruction" as it implies that underprepared students are leveraging one another to succeed in mastering the material. It seems that this also instilled these students with a sense of belonging in the companion course:

> "Yeah, so that made me feel like, "Oh! Maybe I do know a little something, like more than I thought I did." And then, yeah, working with people, that was great. And some people that I met in 41A, like, they're still my friends now and, like, even outside of 41A. But also, we would do physics outside of class together and like study together, so, it kind of made a support system for the people in it."

Peer instruction was common among all students, with 50% of students who perform better than predicted and 60% of students who perform worse than predicted reporting such interactions.

In addition, 25% of students who perform better than predicted took on a leadership role in their group. Some of these students were also coded as "Peer Instruction," but others were not.

Occasionally, students would report group dysfunction: reflecting situations where one or more students would rush ahead on problems without stopping to make sure the rest of their group understood their thought process, leaving some students struggling to even copy down the answers:

*"And we would be in the group doing our, like, work and then, like, they would be, like, flying through the packets and I'd still be on the first or second page. And so it would be hard to ask questions, because you don't want to feel like you're impeding on them doing their work. So that would be the challenge."*

*". . .but everyone in my group had even less, like, physics experience or inclination than me. . .so, like, if I didn't understand it then none of us understood it."*

We coded students who felt like they were being left behind as "Left Behind." This feeling is reported by smaller numbers of students, and is similar across students who perform better than predicted (38%) and worse than predicted (30%). One student, who performed worse than predicted, reported that no one in their group knew what to do at times ("No Direction").

Finally, we note that our prior statistical analyses [6] found no main effect of the companion course, but did find a positive effect for students with the lowest levels of prior preparation.

### Instructional resources

Question 6 in the interview protocol probed what students did to study and prepare for Physics 1 outside of regular homework assignments. This elicited responses that one would expect, such as reading the textbook before class (# 19 in Table 3):

*". . .I would definitely do readings on the weekends before class. I would try to get the readings in before the material would be covered in class, so that I had a sort of understanding of what was going on when I got to class. . ."*

using external/online resources (# 16 in Table 3) such as Khan Academy videos

*"Um, sometimes like Khan Academy just to, like, understand basic stuff. I feel like there's not too much in-depth that you can learn from those videos, or at least if there was a specific question I had, sometimes those videos can't help you. But I think it just overall brushing up a concept that I maybe missed during class, then that's when I would go to online videos."*

or consulting friends/family members who were not in the class

*". . .most of the time I worked on my own, or I would call my dad [a physics teacher] and he would spend a few hours on the phone with me."*

and doing practice problems/exams. All students reported using practice midterm exams provided by the instructors to aid their study, and most students would review or rework completed homework assignments in advance of the exam. Most students reported that they would attempt to re-work the problems and reference the solution when they got stuck, but did not indicate how heavily they relied on the written solutions. Another 18 students reported doing additional practice problems (beyond the practice midterm and homework review; # 18 in Table 3):

*". . .You just have to solve so many problems that you know every type of problem, so when you go to the midterm it's like, "oh, I have this type of problem—I know exactly what to do." And I feel like that's the best way to prepare for science classes in college, because there's so many problems you can write. . ."*

These problems mainly came from the textbook [16] or from companion course worksheets that were not completed in class. We coded this behavior as "Extra Practice," as it indicated going beyond the typical level of study for students in this class.

The use of external resources and reading ahead are reported at similar rates between the two groups of students. 53% of students who perform better than predicted use external resources compared with 58% of students who perform worse than predicted. 33% of students who perform better than predicted read before class compared with 26% of students who perform worse than predicted. We do see more students who perform worse than predicted reporting doing practice textbook problems (58%, compared with 47% of better than predicted students). In addition to these resources, a small number of students who perform worse than predicted reported meeting individually with members of the teaching team (10%).

## Student characteristics & extracurriculars

Most students (*N = 27*) indicated that they were involved in extracurricular activities at some point during the interview (#22 in Table 3). Substantially more students who perform worse than predicted report extracurricular involvement (89% compared with 67% of students who perform better than predicted).

One characteristic that came up during the interviews was that some students took Physics 1 as freshmen, while other students delayed taking the course until their second, third, or fourth years (# 20 in Table 3) 53% of the students who delayed did worse than predicted while 33% did better. Students did not always say why they would delay, but some indicated that they were nervous about taking the course because of its reputation for being difficult.

In addition, a few of the students were NCAA athletes, all of whom performed worse than predicted.

## Student attitudes

None of the questions directly probed typical social psychological constructs such as mindset or imposter syndrome because in previous investigations in this population we had found these factors not to be statistically significant predictors of course performance. Still, a variety of attitudes and mindsets became apparent during analysis of interview transcripts. One notable effect was students' overall change in attitude toward physics as a whole, or their ability to do physics; this was coded as negative (-1), neutral (0) or positive (1). Surprisingly, most students had a netural or positive change in attitude, whether they performed better or worse than predicted. 13% of students who perform better than predicted and 11% of students who perform worse than predicted showed a negative change in attitude. 27% of students who performed better than predicted and 42% of students who perform worse than predicted showed no change in attitudes. 60% of students who perform better than predicted and 47% of students who performed worse than predicted showed positive shifts in their attitude toward physics.

Another theme that we noticed during transcript analysis was student mindset regarding intelligence: whether a student's comments reflected a growth mindset, e.g.

*"I like the math and sciences, but I wasn't like super-pumped-up to take physics to be honest. Um, I knew it would be a difficult class for me, but, yeah, I was ready to learn."*

or a fixed mindset:

*"Getting. . .having like a perfect, like, academic record was, I think, too far integrated into my identity as a person, and I thought when I didn't have that, it would be like I lost a part of myself."*

Relatively few students showed any particular mindset in their interviews as we did not directly ask them about this. However, 20% of students who perform better than predicted and 11% of students who perform worse than predicted showed evidence of a growth mindset. Small and similar percentages of students from both groups exhibited a fixed mindset in their interview.

Some students exhibited "grit"—persevering in STEM or physics 1 in the face of great barriers to their success:

*"Yeah, actually my mindset when I came into [Stanford] was that I wanted to be a physics major because, even though I hated physics in high school and I was not good at it, that sort of gave me motivation to want to be good at it, so I decided to major in it, yeah."*

20% of students who performed better than predicted showed evidence of grit, compared with 11% of students who perform worse than predicted.

Another common experience among the students we interviewed was the feeling that everyone else in the class already knew what was going on and was ahead of them

*"So a lot of my peers, were like, at a much higher level than others. Um, I think that that's what made the experience a little bit hard because I feel. . .I felt like I was always behind, but it wasn't necessarily that I was behind it was just that everyone else in that class had maybe taken AP physics but they were taking physics [1] again"*

We call this a lack of confidence. 32% of students who perform worse than predicted showed this lack of confidence, whereas only 20% of students who performed better than predicted lacked confidence.

A substantial number of students complained about a "disconnect" between the example problems the instructor went over in class and the problems seen on exams and homeworks:

*"I think my biggest problem that I had with it was that I found there to be such a disconnect between the class, the problem sets, and the exams. So, I would go to class and they would teach us Newton's Laws, they'd say: F = ma. And I would say: okay I get this. And then the problem set would be like, I don't. . .I remember the second one was like: someone falls off a bike I think and they hit their head and what's like the deceleration of like their brain into their helmet. And I remember looking at it and saying like what, how am I supposed to solve this with just F = ma."*

Interestingly, more students who performed better than predicted (40%, compared with 26% of students who performed worse than predicted) mentioned this disconnect in their interviews. Another common complaint was that the course moved too fast. We coded complaints about pacing from 53% of students who perform better than predicted and 58% of students who perform worse than predicted.

## Discussion

From the analysis of 34 interviews of students with relatively weak preparation to take physics 1, we identified 15 practices, 10 of which were widespread, and seven attitudes that students felt were important to their experience in physics 1. There are several notable trends in the data.

### Use of instructional resources

Instructors may think that differences in exam performance are related to students' study habits, but our data show a minimal correlation between commonly reported study habits—reading before class, doing extra practice problems, and using external resources—and exam performance. Indeed, we find that students who perform worse than predicted are more likely to do practice textbook problems. This lack of correlation between study habits and performance suggests that, though students are using these commonly-advocated study strategies, the strategies may not be as effective as most physics instructors think. We had more detail about students' study habits than previous works, which only looked at whether the student had a quiet study space [10, 17]. This finding is problematic, as students who performe worse than predicted may have more trouble assessing their own knowledge, and indeed believe that these study practices are helping them when they may not be. Furthermore, we find that all students who reported meeting one-on-one with instructional staff performed worse than predicted on their exams. A plausible explanation for this is that these were the students who were struggling most, but did not receive individual attention until too late in the term and were thus unable to make up for the gaps in knowledge that already existed.

An interesting finding was that most students from both groups reported reworking homework problems (using provided solutions to help them) and practice midterms to study. These also seem like helpful practices, but we were unable to determine in the interviews how heavily students relied upon the given solutions when working or reworking these particular problems. It is plausible that students who perform worse than predicted relied more heavily on the provided solutions, and thus did not fully engage in the process of working these additional problems.

### Out of class interactions

This data suggests that students' interactions outside the classroom is correlated with performance, consistent with some prior research [16]. Student interactions outside the classroom characterized by overreliance on peers, or a general failure to struggle individually with the course material, were more commonly reported by students who perform worse than predicted. Curiously, we do not see the opposite effect of positive interactions being reported more by students who perform better than predicted. One explanation for this could be that positive out-of-class interactions were quite common, and thus taken for granted and so not always mentioned in the interviews. Although instructors have less control over students' out of class study interactions, they can tell them what kinds of such interactions are known to lead to better and worse performance outcomes.

### Student characteristics

There were several student characterisitics that appear to be correlated with performance. Athletes and non-first year students were substantially more likely to perform worse than predicted on the final exam. The result for athletes is unsurprising given the many demands on those students' time, and is consistent with the finding that students who perform worse than

predicted report higher levels of extracurricular involvement. The finding about non-first year students is notable. Though these students have had more time to adapt to the college environment, we suspect they were more likely to be in majors where physics was not a central requirement. For example, computer science is a popular major which requires only one quarter of physics. Many students thus wait to take physics until later in their careers. Because physics is not as important for these majors and being completed only to satisfy a requirement, the students may put a lower priority on the course. Alternatively, these students may have also forgotten more of the high school physics that they learned. However, prior research suggests that students' conceptual understanding of physics is relatively stable over time scales of one year or more [18].

## Student attitudes

1. Though we did not intend to investigate effects of students' attitudes on their course performance, there are some notable findings. First, students who perform better than predicted are more likely to show a growth mindset and exhibit grit. This is in line with previous work showing that mindset is linked to academic performance [9]. We find that a large fraction of all students have a positive change in attitude toward physics, although this is less for the lower-than-predicted students. We also see that, not surprisingly, more students who perform worse than predicted exhibited a lack of confidence. It is not clear whether this is the cause or result of their performance, however. Finally, we found that students who perform better than predicted were more likely to complain about a disconnect between homework and exam problems. We take this to be evidence of students being in a stage of "knowing what they do not know." These students develop more content knowledge and are aware of gaps in that knowledge, whereas the students who perform worse than predicted may not recognize these gaps. Finally, we note that all these attitudinal variables are individual-related rather than course related, so one might expect that students who perform better than predicted in this course also perform better in other courses. We do indeed find that students who perform better than predicted have higher cumulative GPAs than students who perform worse than expected, but this difference is small and not significant ($p = 0.13$, Mann-Whitney U-test).

One notable issue, that is not discussed above, is that the interviewers were typically unable to infer how well a student had performed in Physics 1 or how stressful the course was for them from the tone of the interview. We coded students' overall experience in the course as negative (-1), neutral (0) or positive (1) based on the interviews, but when we then looked at their course performance, we saw no correlation with these ratings of their experience. While this is a crude measure of students' perceptions of the course, this is an unexpected result. We cannot say whether this would be true in other student populations, but it suggests that the thoughts and experiences of students who performed poorly in introductory physics are more varied and nuanced about the course than many instructors may think.

It is interesting that issues of test anxiety and self-efficacy were not apparent from students' interviews, despite other researchers seeing these issues as significant predictors of course performance. It is possible that both of these issues are relevant to student performance, but simply were not elicited from the more general interview questions that we asked. However, the research on impacts of test anxiety and social psychological factors has shown inconsistent results as to how important these factors are. Given that these students were drawn from the same population as in Ref. [6], it seems most likely that these issues are simply not relevant for the students studied.

A limitation of the current study is that we use final exam score as a measure of performance, but our questions to the students were about their experiences over the entire course and not just in preparation for the final exam. A more appropriate measure of course performance might therefore be final course grades, but we note that 40% of the final course grade is determined by the final exam, so there is a strong correlation. Furthermore, none of the students mentioned significant changes in the way that they approached or viewed the course during their interviews, so it seems likely that the behaviors they reported were stable over the course of the term.

## Conclusions

The first, and perhaps most notable conclusion from this study, is the rather small and consistent number of factors that students indicate played a large part in their physics 1 experience and success. There are a large and more idiosyncratic set of factors that students could list as impacting their course performance, such as living situations, personal relationships, health issues, etc., but these students listed only a small number of factors, most of which were widely seen as important. The absence of these other issues may be because students did not feel comfortable bringing up such issues in the interviews, although they spoke very freely about their intense personal highs and (mostly) lows associated with the course, and its impact on their career choices and other important personal decisions. Another reason for the absence may be that students at elite universities such as Stanford face somewhat fewer such barriers to education than the majority of college students.

On the list of factors students noted as important, there were a number of factors that we expected would be correlated with performance that were not, such as types of study habits. Involvement in extracurricular activities was negatively correlated with exam performance, as expected, suggesting some issues related to time management,.

This study represents a small step in answering an outstanding question in physics education research: how can instructors best tailor their instruction to achieve success for all their students, given a wide range of prior knowledge and experience? We offer some suggestions in the non-exhaustive list below.

Our data suggest that some students may struggle to make effective use of instructional resources, including their peers, the textbook, and online resources. Activities that support the use of resources (e.g., practice textbook problems) with explicit instruction as to how to use those resources most effectively for learning are likely to be helpful for students. Research suggests that incorporating instruction on study skills into the introductory physics courses would be a more effective way than general academic support programs to help less prepared students succeed and become self-directed learners in physics [19–23].

Our data also suggest that students' out-of-class interactions may be consequential for their learning. While instructors have limited control over these interactions, there are some steps they can take. For example, instructors can assign students to study groups, run get-to-know-you activities, group students in recitations based on where they live, and host spaces for students to come work on homework in groups with TAs [16]. Providing students with a discussion of the benefits as well as guidance as to how to most effectively learn from one another, may help students obtain greater benefit from such interactions.

Finally, our data also support the growing literature that suggests student mindsets affect academic performance. Instructors should be sure to reinforce a growth mindset in their students through one of the many interventions that exist for such purposes [24]. Instructors should also work to combat imposter syndrome among their students, avoiding students feeling like everyone else in the class knows more than they do. One way to accomplish this would

be to better match the introductory coursework to students' actual levels of prior preparation. We are attempting to address this by creating a separate introductory course for students with little to no physics preparation. This course will cover the same material as the course we studied here, but will be taught using cooperative group problem-solving. Because all of the students start with little physics knowledge and we will explicitly acknowlegdge this, it might help abate imposter syndrome. Furthermore, the active learning methods used have been shown to disproportionately benefit underrepresented students.

We end this article with a reminder to the reader that we are dealing with an unusual student population. Though we selected students in the lowest deciles of incoming preparation in Physics 1 at Stanford as measured by the FMCE, they are likely better prepared for introductory physics than students at many colleges and universities. Still, the issues raised in these interviews may plausibly be relevant to students at any university, and we indeed saw a great degree of consistency across students in our sample. Further research should be carried out to determine whether these issues are unique to Stanford students, or whether these are issues common to introductory physics students more broadly.

## Acknowledgments

The authors thank course instructors and the office of Institutional Research & Decision Support at Stanford University for providing student data. Conversations with Shima Salehi are greatly appreciated. The authors express their deepest gratitude to the students who participated for sharing their experiences with us and making this work possible.

## Author Contributions

**Conceptualization:** Eric Burkholder, Carl Wieman.

**Formal analysis:** Eric Burkholder, Lena Blackmon.

**Investigation:** Eric Burkholder, Lena Blackmon.

**Supervision:** Carl Wieman.

**Writing – original draft:** Eric Burkholder.

**Writing – review & editing:** Eric Burkholder, Carl Wieman.

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
