## [Decision Letter · Decision Letter 0]

16 Oct 2020

PONE-D-20-27321

Hidden variables: predicting student performance in introductory physics

PLOS ONE

Dear Dr. Burkholder,

Thank you for submitting your manuscript to PLOS ONE. After careful consideration, we feel that it may have merit but does not fully meet PLOS ONE’s publication criteria as it currently stands. Therefore, we invite you to submit a revised version of the manuscript that addresses the points raised during the review process. The reviewers have raised concerns regarding statistical methods, inferences, framing of the conclusions and the title of the article.

We look forward to receiving your revised manuscript.

Kind regards,

Mohammed Saqr, Ph.D

Academic Editor

PLOS ONE

Journal Requirements:

Reviewers' comments:

Reviewer's Responses to Questions

**Comments to the Author**

1. Is the manuscript technically sound, and do the data support the conclusions?

Reviewer #1: Partly

Reviewer #2: Yes

Reviewer #3: Partly

2. Has the statistical analysis been performed appropriately and rigorously? 

Reviewer #1: No

Reviewer #2: Yes

Reviewer #3: N/A

3. Have the authors made all data underlying the findings in their manuscript fully available?

Reviewer #1: No

Reviewer #2: No

Reviewer #3: Yes

4. Is the manuscript presented in an intelligible fashion and written in standard English?

Reviewer #1: Yes

Reviewer #2: Yes

Reviewer #3: Yes

5. Review Comments to the Author

Reviewer #1: Reviewer comments for each prompt follow below each prompt.

1 and 2. Is the manuscript technically sound, and do the data support the conclusions? Has the statistical analysis been performed appropriately and rigorously?

The experiments, statistics, and other analyses are performed to a high technical standard and are described in sufficient detail for some of the work but not all. The techniques for collecting the data, identifying factors important to students’s experiences in introductory physics classes and arriving at the percentages in Table III are clearly described. The methods for evaluating the validity of conclusions that could be drawn from the differences in percentages are not provided. These methods seem critical for this data set that involves responses from a small number of participants (N=15 and 19).

The abstract does not represent the findings accurately. The phrase ”such as collaborative interactions between students, were correlated with course performance” is misleading. The discussion of the data and the data analysis as presented do not adequately support drawing any conclusions on the effects on expected course performance. In particular, the abstract does not include a reference to one of the most significant looking influences on expected course performance. The data suggest that students who take physics more than a year after high school seem to perform worse. This difference in performance is as large or larger than any other effect considered in the analysis. The manuscript suggests that this effect may be due to the students being relatively uninterested in physics. Have the authors considered other possibilities such as students who “delay” forgetting more of the physics they learn.

Echoing the above, it is not possible to tell what criteria the authors used to draw conclusions. If there were statistical methods applied, then they should be described.

3. No, according to the author's statement with the manuscript.

4. Is the manuscript presented in an intelligible fashion and written in standard English?

Yes except for a couple of points.

It would be helpful for Table III to include the N’s for the table headings “Better than predicted” and “Worse than predicted”.

It would be helpful for Table III to indicate which of the items had too few instances for the authors to include them in the quantitative analysis. It is confusing to see the percentages associated with them listed as the authors state in the text that they will not include them in the quantitative analysis.

Additional Comments:

The authors acknowledge that students may be reluctant to talk about their experiences in courses in which they feel they have performed poorly. The financial incentive seems like a neutral way to attract participants to the study. In my opinion, the more aggressive tactics to boost the numbers that included enlisting third parties (like diversity offices and department administrators) might feel threatening to the students. It would help if the manuscript included what the guidelines were for soliciting responses.

Reviewer #2: The manuscript is of general interest in education. I am not sure the

conclusion meshes very well with the title of the paper though, because

the hidden variable are not quite and clearly identified as the ones

that affect the student performance. I do see lots of variables, but the

authors should summarize the "few" most important of all the listed

variables as the "Hidden variables that affect student performance," I

think.

Other items:

1) The authors claim 34 students is a large number of students for such study. I am

not sure that 34 is enough for such lofty goals, but the study does involve

much research even with 34 students.

2) Line 30 should cite reference 6

There are a few typos, please check. Some found are on lines:245 (use 'they are'

rather than 'them'); 336 (use 'these'); line 353 (use 'neutral');

427 (use characteristics');

3) Explain on the first apearance what the authors mean by N, see, for example

line 182. It's confusing when one first encounters it.

Reviewer #3: This study tries to identify issues (hidden variables) that may be important for student experience and performance in introductory physics, especially for students weakly prepared for the course. Based on the interviews of 34 students, the authors identified these issues and categorized them into attitudes and practices. They then analyzed the difference between better-than-predicted and worse-than-predicted student groups in each of these issues. Some issues were found to have a negative or positive correlation with course performance: Out-of-class social interactions, extracurriculars, and athletics correlated with worse performance, while certain student characteristics (growth mindset and grit) correlated with better performance. Surprisingly, use of instructional resources showed no or a negative correlation with performance. In my opinion, this study does not reveal insightful hidden variables or provide practical guidelines specific to intro phys to improve student performance. Rather, it reaffirms that variables not specific to intro phys course such as personal attributes or pre-existing conditions (grit, leadership, mindset, gap year, extracurriculars, student athletes, etc.) are the strongest predictors of the course performance.

Major issues:

The interviews are retrospective. Students are more likely to be influenced by their experience in the courses that they are taking currently. It also appears that the 34 students took the intro phys course from different lecturers in different semesters. These variables will definitely confound the interpretation of their interviews.

Overall, better and worse student groups identified most practices at a similar rate. Therefore, these practices might not be good predictors of student performance as the authors suggest. But it is also possible that underperforming students simply “do not know what they do not know” and are misled to think that they increased their knowledge by pursuing such practices.

Since the most important attitudes identified from this study are growth mindset, grit, and confidence which are individual-related rather than course-related, students in the better-than-predicted group are more likely to be better in other courses. Have the authors compared average GPAs of the two groups based on other courses?

The authors use the final exam score as a metric for performance while asking students questions not explicitly related to the final preparation. Student experience over the whole semester does not necessarily correlate with the final exam score, and I think this lack of correlation would be more pronounced among underperforming students.

Minor questions:

It sounds like the one-hour companion course would be a great help for students. Did the authors find any positive effect of taking this course?

One of the questions which I personally deem as most important did not make it to the list of issues: “Would you review solutions or re-work homework problems when studying?” Is it because most students did not mention this practice?

6. PLOS authors have the option to publish the peer review history of their article (what does this mean?). If published, this will include your full peer review and any attached files.

Reviewer #1: No

Reviewer #2: **Yes: **Javier E. Hasbun

Reviewer #3: No

---

## [Author Response · Author response to Decision Letter 0]

20 Oct 2020

We have attached our response to the reviewers with the article. We thank them for their time and consideration, as their thoughts have significantly improved the manuscript.

---

## [Decision Letter · Decision Letter 1]

4 Dec 2020

What factors impact student performance in introductory physics?

PONE-D-20-27321R1

Dear Dr. Burkholder,

We’re pleased to inform you that your manuscript has been judged scientifically suitable for publication and will be formally accepted for publication once it meets all outstanding technical requirements.

Kind regards,

Mohammed Saqr, Ph.D

Academic Editor

PLOS ONE

Additional Editor Comments (optional):

Reviewers' comments:

Reviewer's Responses to Questions

**Comments to the Author**

1. If the authors have adequately addressed your comments raised in a previous round of review and you feel that this manuscript is now acceptable for publication, you may indicate that here to bypass the “Comments to the Author” section, enter your conflict of interest statement in the “Confidential to Editor” section, and submit your "Accept" recommendation.

Reviewer #1: All comments have been addressed

Reviewer #2: All comments have been addressed

Reviewer #4: All comments have been addressed

2. Is the manuscript technically sound, and do the data support the conclusions?

Reviewer #1: Yes

Reviewer #2: Yes

Reviewer #4: (No Response)

3. Has the statistical analysis been performed appropriately and rigorously? 

Reviewer #1: Yes

Reviewer #2: Yes

Reviewer #4: (No Response)

4. Have the authors made all data underlying the findings in their manuscript fully available?

Reviewer #1: No

Reviewer #2: No

Reviewer #4: (No Response)

5. Is the manuscript presented in an intelligible fashion and written in standard English?

Reviewer #1: Yes

Reviewer #2: Yes

Reviewer #4: (No Response)

6. Review Comments to the Author

Reviewer #1: (No Response)

Reviewer #2: Still a few typos left:

line 355 replace 'thse' to 'these'

line 459 replace 'characterisitics' to 'characteristics'

check throughout to make sure there are no typos left

Reviewer #4: (No Response)

7. PLOS authors have the option to publish the peer review history of their article (what does this mean?). If published, this will include your full peer review and any attached files.

Reviewer #1: No

Reviewer #2: **Yes: **Javier E. Hasbun

Reviewer #4: No

---

## [Editor Report · Acceptance letter]

9 Dec 2020

PONE-D-20-27321R1 

 What factors impact student performance in introductory physics? 

Dear Dr. Burkholder:

I'm pleased to inform you that your manuscript has been deemed suitable for publication in PLOS ONE. Congratulations! Your manuscript is now with our production department. 

Kind regards, 

on behalf of

Dr. Mohammed Saqr 

Academic Editor

PLOS ONE